# Masking: A New Perspective of Noisy Supervision

**Bo Han**[*1,2], **Jiangchao Yao**[*3,1], **Gang Niu**[2], **Mingyuan Zhou**[4],
**Ivor W. Tsang**[1], **Ya Zhang**[3], **Masashi Sugiyama**[2,5]

[1]Centre for Artificial Intelligence, University of Technology Sydney
[2]Center for Advanced Intelligence Project, RIKEN
[3]Cooperative Medianet Innovation Center, Shanghai Jiao Tong University
[4]McCombs School of Business, The University of Texas at Austin
[5]Graduate School of Frontier Sciences, University of Tokyo

## Abstract

It is important to learn various types of classifiers given training data with noisy labels. Noisy labels, in the most popular noise model hitherto, are corrupted from ground-truth labels by an unknown *noise transition matrix*. Thus, by estimating this matrix, classifiers can escape from overfitting those noisy labels. However, such estimation is practically difficult, due to either the indirect nature of two-step approaches, or not big enough data to afford end-to-end approaches. In this paper, we propose a human-assisted approach called "*Masking*" that conveys human cognition of *invalid class transitions* and naturally speculates the structure of the noise transition matrix. To this end, we derive a *structure-aware probabilistic model* incorporating a structure prior, and solve the challenges from structure extraction and structure alignment. Thanks to Masking, we only estimate unmasked noise transition probabilities and the burden of estimation is tremendously reduced. We conduct extensive experiments on *CIFAR-10* and *CIFAR-100* with three noise structures as well as the industrial-level *Clothing1M* with agnostic noise structure, and the results show that Masking can improve the robustness of classifiers significantly.

## 1 Introduction

It is always challenging to learn from noisy labels [2, 34, 4, 37, 25], since these labels are systematically corrupted. As a negative effect, noisy labels inevitably degenerate the accuracy of classifiers. This negative effect becomes more prominent for deep learning, since these complex models can fully memorize noisy labels, which correspondingly degenerates their generalization [48]. Unfortunately, noisy labels are ubiquitous and unavoidable in our daily life, such as web queries [24], social-network tagging [6], crowdsourcing [45], medical images [8], and financial analysis [1].

To handle such noisy labels, recent approaches explore three directions mainly. One direction focuses on training only on selected samples, which leverages the sample-selection bias [18] to overcome the label noise issue. For example, MentorNet [19] trains on "small-loss" samples. Meanwhile, Decoupling [26] trains on "disagreement" samples, for which the predictions of two networks disagree. However, since the data for training are selected on the fly rather than selected in the beginning, it is hard to characterize these sample-selection biases, and then it is hard to give any theoretical guarantee on the consistency of learning.

Another direction develops regularization techniques, including explicit and implicit regularizations. This direction employs the regularization bias to overcome the label noise issue. Explicit regulariza-

---

[*]The first two authors (Bo Han and Jiangchao Yao) made equal contributions. The implementation is available at https://github.com/bhanML/Masking.

tion is added to the objective function, such as manifold regularization [5] and virtual adversarial training [30]. Implicit regularization is designed for training algorithms, such as temporal ensembling [20] and mean teacher [41]. Nevertheless, both approaches introduce a permanent regularization bias, and the learned classier barely reaches the optimal performance [9].

The last direction estimates the noise transition matrix without introducing sample-selection bias and regularization bias. As an approximation of real-world corruption, noisy labels are theoretically flipped from the ground-truth labels by an unknown noise transition matrix. In this approach, the accuracy of classifiers can be improved by estimating this matrix accurately. The previous methods for estimating the noise transition matrix can be roughly summarized into two solutions.

One solution estimates the noise transition matrix in advance, and subsequently learns the classifier based on this estimated matrix. For example, Patrini et al. [32] leveraged a two-step solution to estimate the noise transition matrix. The benefit is to require limited data only for the estimation procedure. Nonetheless, their method is too heuristic to estimate the noise transition matrix accurately.

The other solution jointly estimates the noise transition matrix and learns the classifier in an end-to-end framework. For instance, on top of the softmax layer, Sukhbaatar et al. [39] added a constrained linear layer to model the noise transition matrix, while Goldberger et al. [10] added a nonlinear softmax layer. The benefit is the generality of their unified learning framework. However, their brute-force learning leads to inexact estimation due to a finite dataset.

Therefore, an important question is, with a finite dataset, can we leverage a constrained end-to-end model to overcome the above deficiencies? In this paper, we present a human-assisted approach called "*Masking*". Masking conveys human cognition of invalid class transitions (i.e., cat $\nleftrightarrow$ car), and speculates the structure of the noise transition matrix. The structure information can be viewed as a constraint to improve the estimation procedure. Namely, given the structure information, we can focus on estimating the noise transition probability along the structure, which reduces the estimation burden largely.

To instantiate our approach, we derive a structure-aware probabilistic model, by incorporating a structure prior. In the realization, we encounter two practical challenges: structure extraction and structure alignment. Specifically, to address the structure extraction challenge, we propose a tempered sigmoid function to simulate the human cognition on the structure of the noise transition matrix. To address the structure alignment challenge, we propose a variant of Generative Adversarial Networks (GANs) [12] to avoid the difficulty of specifying the explicit distributions. We conduct extensive experiments on two benchmark datasets (*CIFAR-10* and *CIFAR-100*) with three noise structures, and the industrial-level dataset (*Clothing1M*[46]) with agnostic noise structure. The experimental results demonstrate that the proposed approach can improve the robustness of classifiers significantly.

## 2 A new perspective of noisy supervision

In this section, we explore the noisy supervision from a brand new perspective, namely the *structure* of the noise transition matrix. First, we discuss where noisy labels normally come from, and why we can speculate the structure of the noise transition matrix. Then, we present the representative structures of the noise transition matrix.

In practice, noisy labels mainly come from the interaction between humans and tasks, such as social-network tagging and crowdsourcing. Assume that the more complex the interaction is, the more efforts human beings spend. Due to cognitive psychology [29], human cognition can mask invalid class transitions, and highlight valid class transitions automatically. This denotes that, human cognition can speculate the structure of the noise transition matrix correspondingly.

One effort-saving interaction is that, you tag a cluster of scenery images in social networks, such as beach, prairie, and mountain. However, the foreground of some images appears a dog or a cat, yielding noisy labels [35]. Thus, human cognition masks invalid class transitions (i.e., beach $\nleftrightarrow$ mountain), and highlights valid class transitions (i.e., beach $\leftrightarrow$ dog). This noise structure should be the diagonal matrix coupled with two column lines, namely a *column-diagonal* matrix (Figure 1(a)), where the column lines correspond to the dog and cat classes, respectively.

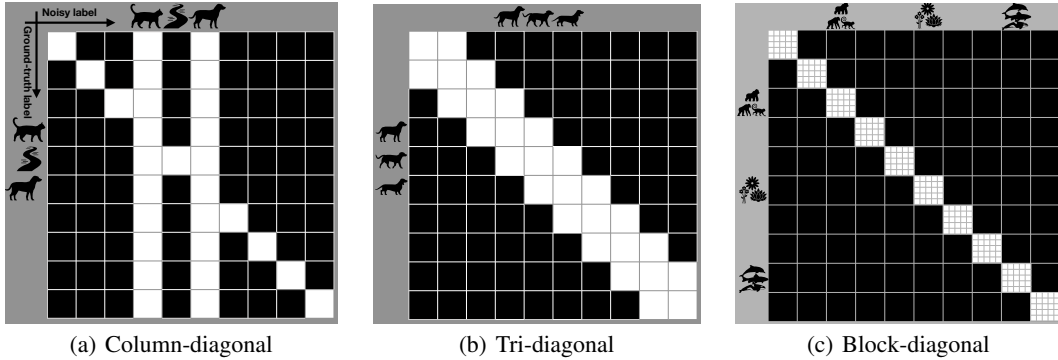

| (a) Column-diagonal | (b) Tri-diagonal | (c) Block-diagonal |

Figure 1: Three types of noise structure. Vertical axis denotes the class of ground-truth label, while horizontal axis denotes the class of noisy label. White block means the valid class transitions, while black block means the invalid class transitions.

Another effort-consuming interactions stem from the task annotation on Amazon Mechanical Turk [2]. Even with high-degree efforts, amateur workers may be potentially confused by very similar classes to yield noisy labels, due to their limited expertise. There are two practical cases.

The fine-grained case denotes that, the transition from one class to its similar class is continuous (e.g., Australian terrier, Norfolk terrier, Norwich terrier, Irish terrier, and Scotch terrier), and workers make mistakes in the adjacent positions [7]. Thus, human cognition can mask invalid class transitions (i.e., Australian terrier ↮ Norwich terrier), and highlight valid class transitions (i.e., Norfolk terrier ↔ Norwich terrier ↔ Irish terrier). This noise structure should be a *tri-diagonal* matrix (Figure 1(b)).

The hierarchical-grained case denotes that, the transition among super-classes is discrete (e.g., aquatic mammals and flowers) and impossible, while the transition among sub-classes is continuous (e.g., aquatic mammals contain beaver, dolphin, otter, seal, and whale) and possible [32]. Thus, human cognition can mask invalid class transitions (i.e., aquatic mammals ↮ flowers), and highlight valid class transitions (i.e., beaver ↔ dolphin). This noise structure should be a *block-diagonal* matrix (Figure 1(c)), where each block represents a super-class.

When we already know the noise structure from human cognition, we only focus on estimating the noise transition probability. However, how can we instill the structure information into the estimation procedure? We are going to answer this question in the following sections.

## 3 Learning with Masking

We briefly show how benchmark models handle noisy labels, and reveal their deficiencies (Section 3.1). Then, we show why the Masking approach can solve such issues (Section 3.2). After the Masking procedure, we present a straightforward idea to incorporate the structure information, and show its potential dilemma (Section 3.2.1). Fortunately, we find a suitable approach to incorporate the structure information into an end-to-end model (Section 3.2.2). To realize this approach, we encounter two practical challenges, and present principled solutions (Section 3.2.3).

### 3.1 Deficiency of benchmark models

Figure 2(a) is a basic model to train a classifier in the setting of noisy labels. Assuming that $x$ represents a "Dog" image, its latent ground-truth label $y$ should be a "Dog" class. However, its annotated label $\tilde{y}$ belongs to the "Cat" class. In essence, the noisy label $\tilde{y}$ is flipped from the ground-truth label $y$ by an unknown noise transition matrix. Therefore, current techniques tend to improve the accuracy of classifier ($x \rightarrow y$) by estimating the noise transition matrix ($y \rightarrow \tilde{y}$).

Here, we introduce two benchmark realizations of Figure 2(a). The first benchmark model comes from Patrini et al. [32], which uses the anchor set condition [21] to independently estimate the noise

[2] https://www.mturk.com/

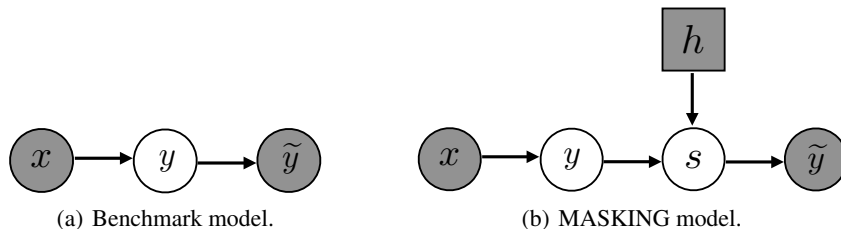

(a) Benchmark model.　　　　　(b) MASKING model.

Figure 2: Comparison of benchmark models and our MASKING model. Assume that, $(x, \tilde{y})$ denotes the instance with the noisy label, and $y$ represents the latent ground-truth label. $T$ is the noise transition matrix, where $T_{ij} = \Pr(\tilde{y} = e^j | y = e^i)$. Left Panel: a benchmark model. Right Panel: MASKING models the matrix $T$ by an explicit variable $s$. Thus, we embed a structure constraint on the variable $s$, where the structure information come from human cognition $h$.

transition matrix ($y \rightarrow \tilde{y}$). Based on the estimated matrix, they learn the classifier ($x \rightarrow y$) by the strategy of loss corrections. However, the estimation phase is not justified for agnostic noisy data, which thus limits the performance of the classifier. The other benchmark model comes from Goldberger and Ben-Reuven [10], which unifies the two steps of the first benchmark model into a joint fashion. Specifically, the noise transition matrix ($y \rightarrow \tilde{y}$) modeled by a nonlinear softmax layer connects the classifier ($x \rightarrow y$) with noisy labels $\tilde{y}$ for the end-to-end training. However, due to a finite dataset, this brute-force estimation may suffer from many local minima.

## 3.2 Does structure matter?

Therefore, given a finite dataset, can we leverage a constrained end-to-end model to solve the above deficiencies? The answer is affirmative. The reason can be explained intuitively: when human cognition masks the invalid class transitions (i.e., cat $\nleftrightarrow$ car), the structure information is available as the constraint. The constrained end-to-end model only focuses on estimating the noise transition probability. The estimation burden will be largely reduced, and thus the brute-force estimation can find a good local minimum more easily.

In this paper, we summarize this human-assisted approach as "*Masking*". Intuitively, with high probability, Masking means some class transitions will not happen (i.e., cat $\nleftrightarrow$ car), while some class transitions will happen (i.e., beaver $\leftrightarrow$ dolphin). Before delving into our realization, we first present a straightforward idea to incorporate the structure information into an end-to-end model as follows.

### 3.2.1 Straightforward dilemma

For structure instillation, the most straightforward idea is to add a regularizer on the objective of an end-to-end model. Namely, we can use a regularizer to represent the structure constraint. Due to the regularization effect in the optimization [11], the noise transition matrix learned by previous models [32, 10] will satisfy our expectation regarding the structure. For example, we can apply the Lagrange multiplier in the benchmark model from [10] to instantiate this idea.

However, such a deterministic method may not be easily implemented in practice due to three reasons. First, such a class of regularizers requires a suitable distance measure to compute the distance between the learned transition matrix and the prior, and the corresponding regularization parameter. In deep learning scenarios, it is quite hard to justify the choice of a distance measure, e.g., why choosing $L_2$ distance instead of other distance measures for the structure instillation. Second, if we leverage a noisy validation set, we need to construct the unbiased risk estimator for the backward correction. This is impossible, as the inverse of estimated noise transition matrix cannot be accurately computed.

Last but not least, even though we construct a clean validation set, it needs to repeat the training procedure to tune the regularization parameter, which consumes remarkable computational resources. Specifically, it requires a lot of non-trivial trials in the training-validation phase to find an optimal weight. For example, the Residual-52 net will take about one week to be well trained on the WebVision dataset, consisting of 16 million images with noisy labels [21]. If we consider adding a

regularizer to instill structure information, each trial for adjusting the weight is a disaster. To sum up, we do not consider this straightforward regularization approach.

### 3.2.2 When structure meets generative model

Based on the previous discussions, we conjecture that the Bayesian method should be a more suitable tool to model the structure information, since the structure information can be explicitly represented as the prior. Following this conjecture, we deduce an end-to-end probabilistic model to incorporate the structure information as shown in Figure 2(b).

Compared to benchmark models in Figure 2(a), we model the noise transition matrix with a random variable $s$, and we instill the structure information by controlling the prior of its corresponding structure variable, termed as $s_o$. Here, we assume there exists a deterministic function $f(\cdot)$ such that $s_o = f(s)$. The reason that we make such an assumption can be intuitively explained with the observation, once one arbitrary matrix is given, human cognition can certainly describe its structure, e.g., diagonal or tri-diagonal. Thus, there must be a function that can implement the mapping from $s$ to its structure $s_o$. For clarity, we present an exemplar generative process for "multi-class" & "single-label" classification problem with noisy supervision.

- The latent ground-truth label $y \sim P(y|x)$, where $P(y|x)$ is a Categorical distribution [3].
- The noise transition matrix $s \sim P(s)$ and its structure $s_o \sim P(s_o)$, where $P(s)$ is an implicit distribution modeled by neural networks without the exact form (i.e., multi-Dirac distribution), $P(s_o) = P(s)\frac{ds}{ds_o}\big|_{s_o=f(s)}$, and $f(\cdot)$ is the mapping function from $s$ to $s_o$.
- The noisy label $\tilde{y} \sim P(\tilde{y}|y, s)$, where $P(\tilde{y}|y, s)$ models the transition from $y$ to $\tilde{y}$ given $s$.

According to the above generative process, we can deduce the following evidence lower bound (ELBO) (the details in Appendix A) to approximate the log-likelihood of the noisy data, which can be named as MASKING. MASKING is a structure-aware probabilistic model:

$$\ln P(\tilde{y}|x) \geq \mathbb{E}_{Q(s)}\left[\ln \underbrace{\sum_y P(\tilde{y}|y, s)P(y|x)}_{\text{previous model}} - \ln \left(Q(s_o)/\underbrace{P(s_o)}_{\text{structure prior}}\right)\bigg|_{s_o=f(s)}\right], \quad (1)$$

where $Q(s)$ is the variational distribution to approximate the posterior of the noise transition matrix $s$, and $Q(s_o) = Q(s)\frac{ds}{ds_o}\big|_{s_o=f(s)}$ is the corresponding variational distribution of the structure $s_o$. Eq. (1) seamlessly unifies previous models and structure instillation, remarked as the following benefits.

**Remark 1** The first term inside the expectation in Eq. (1) recovers the previous benchmark models, representing the log-likelihood from $x$ to the noisy label $\tilde{y}$. The second term inside the expectation in Eq. (1) is for structure instillation, reflecting the inconsistency between the distribution $Q(s_o)$ learned from the training data and the structure prior $P(s_o)$ provided by human cognition.

As a whole, MASKING model benefits from the human guidance (the second term) in the procedure of learning with noisy supervisions (the first term), which avoids the unexpected local minima with incorrect structures in previous works. Moreover, we avoid the difficulty of hyperparameter selection by deducing a Bayesian framework, which does not require the regularization parameter to be tuned.

Note that human cognition may introduce uncertainty. In this case, we just focus on the certain knowledge and use this knowledge as the prior; on the other hand, we dispose the uncertain knowledge by Masking. A special case is when we do not have any transition knowledge, we can unmask the whole matrix, i.e., allowing all possible transitions. This naturally degenerates our MASKING model to the unconstrained S-adapation method.

### 3.2.3 Towards principled realization

To realize the MASKING model, concretely, the second term in Eq. (1), we encounter two practical challenges, and present the principled solutions as follows.

**Challenge from structure extraction:** One challenge comes from how to specify the mapping function $f(\cdot)$ in Eq. (1), which extracts the structure variable $s_o$ from the variable $s$. Without this step, we cannot compute the structure variable $s_o$ for $Q(s_o) = Q(s)\frac{ds}{ds_o}\big|_{s_o=f(s)}$, and let alone optimize the second term in Eq. (1). However, to the best of our knowledge, there is no related work that specifies $f(\cdot)$ in the area of structure extraction.

Here, we explore a principled solution by simulating human cognition on the structure of the noise transition matrix. In terms of the noise transition probability between two classes, human cognition considers the small value (i.e., $0.5\%$) as a sign of the invalid class transition, but the large value (i.e., $20\%$) as a noise pattern [14]. It indicates that, the larger transition probability is favored when quantifying the noise transition matrix into a structure. Such a procedure is very similar to the thresholding binarization operation with a tempered sigmoid function (Eq. (2)). Thus, we can use the following tempered sigmoid function as $f(\cdot)$ to simulate the mapping from $s$ to $s_o$,

$$f(s) = \frac{1}{1 + \exp(-\frac{s-\alpha}{\beta})}, \quad \text{where } \alpha \in (0,1), \beta \ll 1, \tag{2}$$

and we name its output $f(s)$ as the masked structure $s_o$ from $s$. By controlling the location parameter $\alpha$ and the scale parameter $\beta$, Eq. (2) quantifies $s$ into the structure $s_o$ for the second term in Eq. (1).

**Challenge from structure alignment:** The second term in Eq. (1) is to make the structure learned from the training data (represented by $Q(s_o)$), close to the human prior (represented by $P(s_o)$). We consider it as structure alignment. The challenge here is how to specify the distributions $P(s_o)$ and $Q(s_o)$ in Eq. (1) to reasonably measure their divergence for the structure alignment. The smaller divergence between $P(s_o)$ and $Q(s_o)$, the more similar they are.

This question is difficult because, for prior $P(s_o)$, we usually provide one or a few limited structure candidates, and human cognition has the sparse empirical certainty [13] according to a specific noisy data. That is to say, $P(s_o)$ should be a distribution that concentrates on one or a few points, e.g., a multi-Dirac distribution or a multi-spike distribution[4]. These distributions are quite unstable for optimization, since they are approximately discrete, and easy to cause the computational overflow problem [36]. Regarding $Q(s_o)$, it is equal to specifying $Q(s)$ since they are correlated by $s_o = f(s)$. If we find $Q(s)$ such that $\mathbb{E}_{Q(s)}[\ln Q(s_o)/P(s_o)]$ is analytically computable, we can avoid this challenge. However, such a specification with existing distributions is usually intractable.

Fortunately, we can employ the implicit models to deal with the above dilemma [42]. This is because in the implicit models, the distribution is directly simulated with neural networks plus a random noise, which avoids to specify an explicit distribution. Following this methodology, $Q(s)$, $Q(s_o)$ and $P(s_o)$ in Eq. (1) can be implemented like Generative Adversarial Networks (GANs). Specifically, $Q(s)$ and the divergence between $Q(s_o)$ and $P(s_o)$ are parameterized with two neural networks, i.e., one generator and one discriminator, and then play an adversarial game [12]. However, different from the original GANs, our model has one extra discriminator (called as reconstructor), since the first term in Eq. (1) involves $s$, which will act as the second discriminator during the game.

Concretely, the corresponding implementation of each term in Eq. (1) is illustrated in Figure 3, consisting of three modules, generator, discriminator and reconstructor. The generator is responsible for generating a distribution $Q(s)$ of the noise transition matrix $s$, which serves for both terms in Eq. (1). The discriminator implements the function of the second term in Eq. (1) to measure the difference $\mathcal{M}(s_o, \hat{s}_o)$ between the extracted structure $s_o$ with Eq. (2) and our prior structure $\hat{s}_o$. The reconstructor is for the first term in Eq. (1), facilitating the classifier prediction $P(y|x)$ and the noise transition matrix $s$ to yield noisy labels $\tilde{y}$. In this way, we instill the structure information from human cognition into an end-to-end model.

## 4 Related literature

Except several works mentioned before, we survey other solutions for noisy labels here. Statistical learning focuses on theoretical guarantees, consisting of three directions - surrogate losses, noise rate estimation and probabilistic modeling. For example, in the surrogate losses, Natarajan et al. [31] proposed an unbiased estimator to provide the noise corrected loss approach. Masnadi-Shirazi et al.

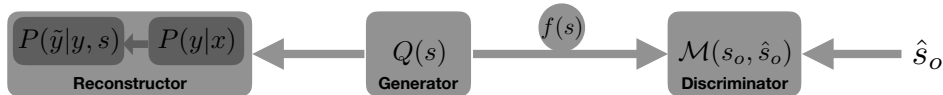

Figure 3: A GAN-like structure to model the structure instillation on learning with noisy supervision.

[27] presented a robust non-convex loss, which is the special case in a family of robust losses [16]. In noise rate estimation, both Menon et al. [28] and Liu et al. [23] proposed a class-probability estimator using order statistics on the range of scores. Sanderson et al. [38] presented the same estimator using the slope of the ROC curve. In probabilistic modeling, Raykar et al. [33] proposed a two-coin model to handle noisy labels from multiple annotators. Yan et al. [47] extended this two-coin model by setting the dynamic flipping probability associated with samples.

Deep learning acquires better performance due to its complex nonlinearity [19, 44, 17, 49]. For example, Li et al. proposed a unified framework to distill the knowledge from clean labels and knowledge graph [22], which can be exploited to learn a better model from noisy labels. Veit et al. trained a label cleaning network by a small set of clean labels, and used this network to reduce the noise in large-scale noisy labels [43]. Tanaka et al. presented a joint optimization framework to learn parameters and estimate true labels simultaneously [40]. Ren et al. leveraged an additional validation set to adaptively assign weights to training examples in every iteration [35]. Rodrigues et al. added a crowd layer after the output layer for noisy labels from multiple annotators [37]. However, all methods require extra resources (e.g., knowledge graph or validation set) or more complex networks.

## 5 Experiments

In this section, we verify the robustness of MASKING from two folds. First, we conduct experiments on two benchmark datasets with three types of noise structure: namely (1) column-diagonal (Figure 1(a)); (2) tri-diagonal (Figure 1(b)); and (3) block-diagonal (Figure 1(c)). Second, we conduct experiments on one industrial-level dataset with agnostic noise structure.

**Benchmark datasets.** *CIFAR-10* and *CIFAR-100* datasets are used. Both datasets consist of 50k samples for training and 10k samples for testing, where each sample is a $32 \times 32$ color image and its label. For *CIFAR-10*, we randomly flip the labels of the training set according to the first two types of noise structure, which has been illustrated in Figure 1 with predefined transition probabilities. For *CIFAR-100*, we implement the similar procedure to generate the noisy data, but follow the last type of noise structure in Figure 1(c). All predefined transition probabilities are found in Table 2 (4th row).

**Industrial-level dataset.** An industrial-level dataset called *Clothing1M* [46] from online shopping websites (i.e., Taobao.com) is used here, where the ground-truth transition matrix is not available. *Clothing1M* includes mislabeled images of different clothes, such as hoodie, jacket and windbreaker. This dataset consist of 1000k samples for training and 1k samples for testing, where each sample is a $256 \times 256$ color image and its label. Although we cannot know the accurate structure prior for *Clothing1M*, we can distill an approximated structure from the pre-estimated transition matrix [46]. We consider the transition probabilities greater than $0.1$ as the valid transition patterns, and the transition probabilities smaller than $0.01$ as the invalid transition patterns.

**Baselines and measurements.** We compare MASKING with the state-of-the-art & the most related techniques for noisy supervision: (1) forward correction [32] (F-correction) and (2) S-adaptation [10]. We also compare it with directly training deep networks on noisy data (marked as (3) NOISY) and clean data (marked as (4) CLEAN). The performance of CLEAN can be viewed as an oracle or an upper bound. The prediction accuracy is used to evaluate the classification performance of each model in the test set. Besides, we qualitatively visualize the noise transition matrix when the training of each model converges, to analyze whether the true noise transition matrix is approached.

**Implementations.** All experiments are conducted on a NVIDIA TITAN GPU, and all methods are implemented by Tensorflow. We adopt the same base network as the classifier of all methods, and apply the cross-entropy loss for noisy labels. The stochastic gradient descent optimizer has been used to update the parameter of baselines and the classifier in MASKING. For the generator and the

discriminator, we follow the advice in Gulrajani et al. [15] to choose the RMSProp optimizer. For both datasets, the batch size is set to 128 for 15,000 iterations. $\alpha$ and $\beta$ in Eq. (2) are respectively set 0.05 and 0.005. The estimation of the noise transition matrix in F-correction and the initialization of the adaption layer in S-adaptation follow the strategy in Patrini et al. [32]. Note that we have tried the original initialization way in Goldberger and Ben-Reuven [10], but it is not better than the way in [32]. More details about the network architectures and the learning rates are summarized in Appendix B. Our implementation of MASKING is available at `https://github.com/bhanML/Masking`.

**Empirical results.** We train MASKING and baselines (except CLEAN) on the noisy datasets and validate the performance in the clean test datasets. Figure 4 depicts the test accuracy on benchmark datasets with three types of noise transition matrix. According to the comparison, we can find that MASKING persistently outperforms F-correction, S-adaptation and NOISY. In terms of tri-diagonal and block-diagonal cases, it almost achieves the performance comparable to that of CLEAN.

Besides, Table 2 presents the visualization of the noise transition matrix estimated by F-correction, S-adaptation and MASKING, as well as the true noise transition matrix. As can be seen, with the guidance of the prior structure, MASKING infers the noise transition matrix better than two baselines. Specifically, we observe that for the estimation on the tri-diagonal structure, both F-correction and S-adaptation severely fail. F-correction pre-estimates a non-ideal matrix, and S-adaptation tunes it worse, while MASKING learns it better. To avoid the performance drop when directly training on the noisy dataset as [19, 3], we have configured the dropout layer in deep neural networks as [3].

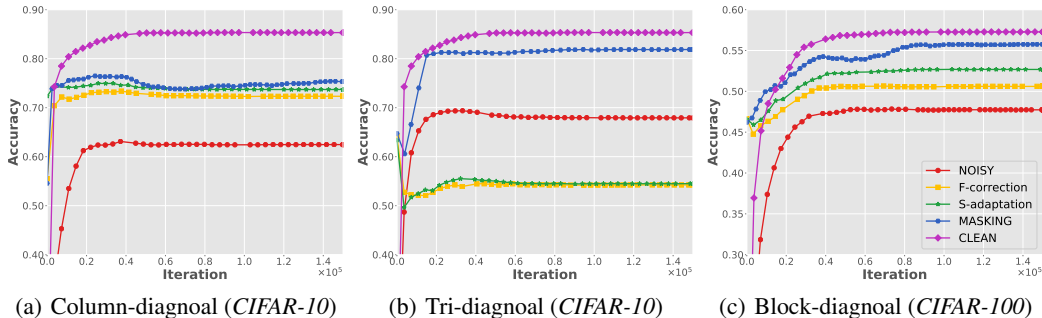

(a) Column-diagnoal (*CIFAR-10*)    (b) Tri-diagnoal (*CIFAR-10*)    (c) Block-diagnoal (*CIFAR-100*)

Figure 4: Test accuracy vs iterations on benchmark datasets with three types of noise structure.

Table 1: Test accuracy on *Clothing1M* with agnostic noise structure.

| Models | Performance(%) |
|---|---|
| NOISY | 68.9 |
| F-correction | 69.8 |
| S-adaptation | 70.3 |
| MASKING | 71.1 |
| CLEAN | 75.2 |

Furthermore, test accuracy of all methods on *Clothing1M* dataset is shown in Table 1. The comparison denotes that, when the noise model of the training data is completely unknown to all methods, MASKING still outperforms other methods. Compared to results in Figure 4, the robustness of MASKING marginally outperforms that of F-correction and S-adaptation. We conjecture two reasons exist: First, the structure prior we used here is not the ground-truth noise structure. Second, the ground-truth noise structure of *Clothing1M* is too complex to be estimated easily. To solve the estimation issue, we can use crowdsourcing to provide labels in practice, and invite an expert to find the accurate noise structure. This should be much cheaper than letting the expert assign labels.

## 6 Conclusions

This paper presents a Masking approach. This approach conveys human cognition of invalid class transitions, and speculates the structure of the noise transition matrix. Given the structure information,

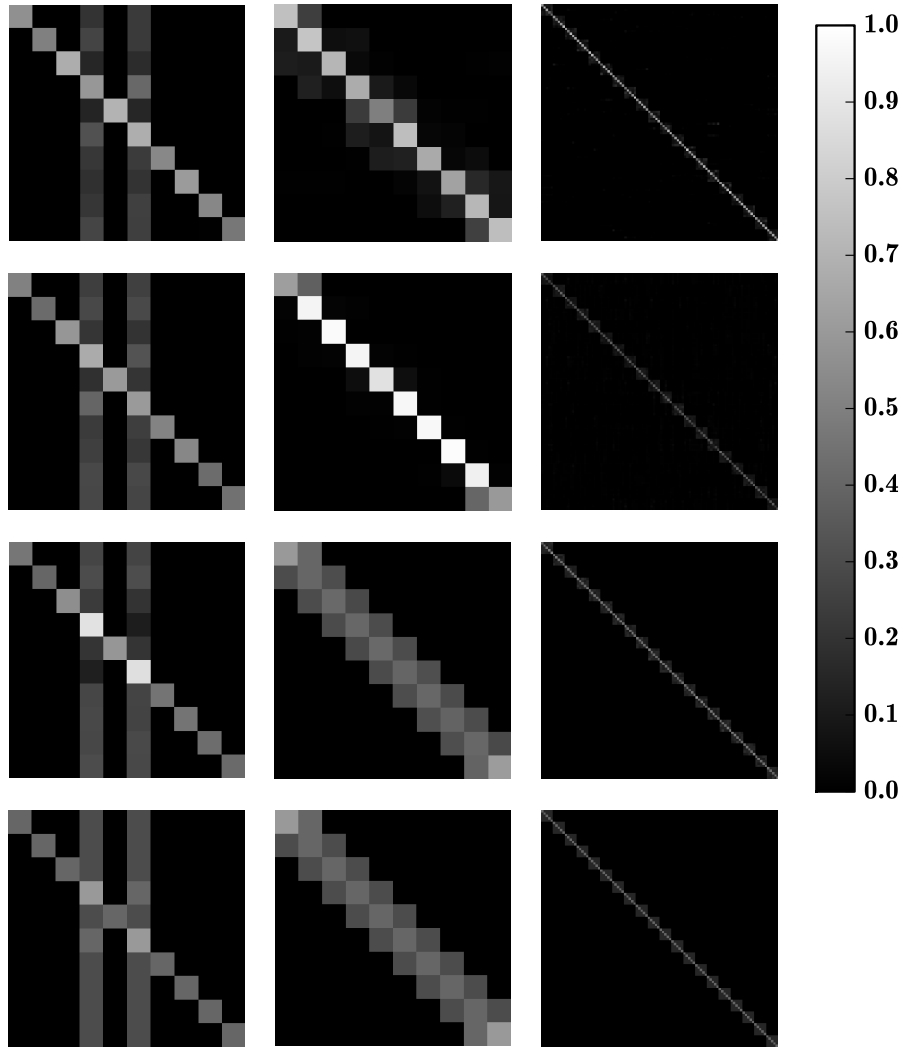

Table 2: The estimation of the noise transition matrix by F-correction (1st row), S-adaptation (2nd row) and MASKING (3rd row), and the truth (4th row) in the case of three types of noise transition structure: column-diagonal (1st column), tri-diagonal (2nd column), block-diagonal (3rd column).

we derive a structure-aware probabilistic model (MASKING), which incorporates a structure prior. Empirical results demonstrate that our approach can improve the robustness of classifiers obviously. In future, we will explore how MASKING self-corrects the incorrect noise structure. Namely, when the noise structure is wrongly set at the initial stage, how does our model correct the initial structure by learning from the finite dataset?

**Acknowledgments.**

MS was supported by the International Research Center for Neurointelligence (WPI-IRCN) at The University of Tokyo Institutes for Advanced Study. IWT was supported by ARC FT130100746, DP180100106 and LP150100671. MZ acknowledges the support of Award IIS-1812699 from the U.S. National Science Foundation. YZ was supported by the High Technology Research and Development Program of China (2015AA015801), NSFC (61521062), and STCSM (18DZ2270700). BH would like to thank the financial support from RIKEN-AIP. JY would like to thank the financial support SJTU-CMIC and UTS-CAI. We gratefully acknowledge the support of NVIDIA Corporation with the donation of the Titan Xp GPU used for this research.

## Footnotes

[3]For the single-label classification, a Categorical distribution is a natural choice. For the multi-label classification, a Multinomial distribution is used, since one example corresponds to multiple labels.

[4]https://en.wikipedia.org/wiki/Dirac_delta_function; https://en.wikipedia.org/wiki/Spike-and-slab_variable_selection

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
