[Supplementary Material]

## A   Deduce ELBO

The objective of our model can be deduced in the perspective of variational inference. Concretely, we can introduce a variational distribution $Q(s)$ to approximate the posterior of noise transition matrix $s$, and apply the Jensen's Inequality to the log-likelihood of data as follows,

$$
\begin{aligned}
\ln P(\tilde{y}|x) &= \ln \int_s \sum_y P(\tilde{y}|y,s)P(y|x)P(s)ds \\
&= \ln \int_s \sum_y P(\tilde{y}|y,s)P(y|x)\frac{Q(s)P(s)}{Q(s)}ds \\
&\geq \mathbb{E}_{Q(s)}\left[\ln \sum_y P(\tilde{y}|y,s)P(y|x) - \ln \frac{Q(s)}{P(s)}\right] \\
&= \mathbb{E}_{Q(s)}\left[\ln \sum_y P(\tilde{y}|y,s)P(y|x) - \ln \frac{Q(s_o)\frac{ds_o}{ds}}{P(s_o)\frac{ds_o}{ds}}\bigg|_{s_o=f(s)}\right] \\
&= \mathbb{E}_{Q(s)}\left[\ln \sum_y P(\tilde{y}|y,s)P(y|x) - \ln\left(Q(s_o)/P(s_o)\right)\bigg|_{s_o=f(s)}\right],
\end{aligned} \tag{3}
$$

where $f(\cdot)$ is the mapping function to transform the noise transition matrix $s$ into its structure $s_o$.

## B   Network structures and training settings

As we have explained in the paper, our GAN-like structure consists of three modules, generator, discriminator and reconstructor. We respectively illustrate their network configuration as follows.

Figure 5: The network configuration of the generator module.

Figure 6: The network configuration of the discriminator module.

The learning rate is initialized as $0.1$, and decreases with the operation *tf.train.exponential_decay* with the staircase in tensorflow. The corresponding decay factor is set $0.1$. The learning rate for generator and discriminator is fixed with $3e - 4$.

| [5,5,3,64] | [1,1,3,3] | [5,5,64,64] | [1,1,3,3] | 384-D | 192-D | | N-D |
|---|---|---|---|---|---|---|---|
| conv | max pooling | conv | max pooling | fc-layer | fc-layer | dropout | fc-layer |

Figure 7: The network configuration of the reconstructor module.