[Reviews · NeurIPS 2018]

Reviewer 1



In real applications, labels usually appear with noise and thus classifiers learned from these noisy labels could not perform well. This paper assumes that the noisy labels are corrupted from ground-truth labels by an unknown noise transition matrix and attempts to learn good classifiers under noisy labels by estimating this transition matrix. Different from the previous works that learn either by a heuristic trick or the brutal-force learning by deep networks, this work incorporates human cognition where invalid class transitions will be masked and valid class transitions will be highlighted. Therefore, the structure of the noise transition matrix can be extracted and aligned to the human's prior to render the transition matrix more effective. Experimental results shows the effectiveness of the proposed approach. Here are my comments: Quality The technical contents in this paper appears clear and sound. Specifically, Equation (1) gives the mathematical objective that combines the previous model ELBO and the idea of this paper---the structure instillation. Then a clear illustration (Figure 3) of the implementation is exhibited. Both of the equation and the illustration of implementation give us a clear notion that they instill the structure information from human cognition into the learned model. The experimental results validate the effectiveness of the proposed approach and are consistent with the three noise structures mentioned in Section 2, which makes the result convincing. Clarity The idea proposed by this paper is distinct that they want to use the human cognition as priors. Every part of this paper is necessary and shows strong connections with each other. For example, in Section 3, the authors first show deficiencies of the previous benchmark models. Then they ask a question that does structure matter. Lead by this question, a series of answers and questions are provided, which guides the readers step by step to the final solution. A suggestion is that Figure 1 should provide an explicit elaboration on the meaning of the axes. The authors should also explain what the white blocks and black blocks mean. In this way, Section 2 will be more understandable. Besides, as far as I am concerned, invalid class transitions refer to those unlikely class transitions by human cognition and mask means the transitions will not happen. I hope the authors could explain these concepts more explicitly. Novelty and Significance Learning under noisy labels has been studied in the last several decades. As mentioned in the paper, there are three directions to tackle this problem. Training on selected samples could introduce the accumulated error due to the sample-selection bias; regularization techniques would bring a regularization bias and thus could not reach the optimal classifier; though assuming the noise is brought by a noise transition matrix and estimating the transition matrix could avoid the deficiency of sample-selection bias or regularization bias, previous efforts hardly estimate the transition matrix accurately. This paper presents a human-assisted approach to incorporate human cognition of easily recognizing the invalid and valid class transitions, which is able to improve the performance notably as well as novel compared to other works estimating the noise transition matrix. Due to the pervasive situation where noisy labels are inevitable, enhancing the performance in this task is very essential. The effective method proposed by this paper that incorporates the human knowledge into the end-to-end framework is a good direction to promote the development of this field. A problem is that what if the priors provided by humans is wrong. This paper does not give an answer. Nevertheless, fortunately, the authors bear this problem in mind and give us a vision on self-correcting which is worth expecting. Overall, I think this paper is a good work to be accepted to NIPS. Thanks for the authors' feedback. The authors provide a solution to the problem where if the prior is not right or not provided. I think this is a good direction and hope to see the corresponding paper in the future.

Reviewer 2



Summary: The paper proposed a way to estimate noise transition probabilities for classification using data with label noise. Specifically, the paper introduces a way to instill human cognition prior that specifies the valid and invalid transition between classes via variational learning formulation. The final algorithm involves a discriminator that takes the proposed (by proposal distribution Q) and pre-defined noise transition matrices and train Q to reduce dis-similarity between two transition matrices. Strength: - The paper proposes a technically sound way to incorporate human cognition prior on the noise transition matrix. - The proposed algorithm learns a correct noise transition matrices for three artificially generated noise transition matrix structures. Weakness: - Although authors try to explain the STABLE model in a Bayesian framework using variational lower bound, as we see in the final form of the algorithm, it doesn't seem different from a simple regularization method illustrated in Section 3.2.1 when discriminator in Figure 3 corresponds to L2 distance. - The experimental protocol seems favorable to the proposed method and unfair to previous methods such as loss correction or S-adaptation. For example, one may inject noise transition prior on S-adaptation by setting corresponding \theta values to zero and renormalize after each update. - In addition, as illustrated in 3.2.1, one can easily include regularization to train noise transition matrix to follow the prior. It doesn't seem like a valid excuse to exclude regularization method from comparison due to the difficulty of experimentation when the proposed method may also involve some hyperparameters that balances training between reconstructor, generator and discriminator. - Experimental validation is not satisfactory as it is only evaluated with artificially generated noise transition constraints.

Reviewer 3



Paper summary ---------------- Learning classifiers under noisy labels is one important issue because of their ubiquities. Since noisy labels are corrupted from ground-truth labels by an unknown noise transition matrix, the paper proposes to estimate the matrix with human assistance. Based on human cognition for invalid class transitions, the paper proposes to speculate the structure of the noise transition matrix. The idea is to derive a structure-aware probabilistic model, which incorporates a structure prior. Technically, the paper addresses structure extraction and alignment. Experiments validate the improvement of robustness of classifiers. Paper summary ---------------- The paper is well motivated by carefully figuring out the major issues and making noisy label learning more practical. The proposed method is reasonable and makes sense, which is interesting as well. Theoretical support is also given. Experimental results are good enough. Finally, the paper is well organized. There are some concerns for the paper. 1. In the part of structure extraction, the paper focuses on a general model which is good. However, it is not intuitive. More intuitive examples or illustrations are needed. 2. The paper introduces human cognition. That would also bring uncertainty simultaneously. How to fix this problem? ------------- The author response is fine for me. I'd like to keep my evaluation the paper.